# Is it time to change the approach of mental health stigma campaigns? An experimental investigation of the effect of campaign wording on stigma and help-seeking intentions

Cassie M. Hazell[1]*, Alison Fixsen[1], Clio Berry[2]

1 School of Social Sciences, University of Westminster, London, United Kingdom, 2 Brighton and Sussex Medical School and School of Psychology, University of Sussex, Falmer, Brighton, United Kingdom

* c.hazell@westminster.ac.uk

**Data Availability Statement:** All relevant data are within the manuscript and its Supporting Information files.

## Abstract

### Introduction

Mental health stigma causes a range of diverse and serious negative sequelae. Anti-stigma campaigns have largely aligned with medical theories and categorical approaches. Such campaigns have produced some improvements, but mental health stigma is still prevalent. The effect of alternative theoretical perspectives on mental health within anti-stigma campaigns has not been tested. Moreover, we do not know their effect on help-seeking intentions.

### Methods

We conducted an online experimental pre-post study comparing the effects of two anti-stigma campaign posters on mental health stigma and help-seeking intentions. One poster adhered to the medical, categorical approach to mental health, whereas the other poster portrayed mental health problems in line with a non-categorical, continuous perspective.

### Results

After controlling for familiarity with the campaign poster, country of residence and pre-test scores, we found no significant between-group differences in terms of help-seeking intentions and all stigma attitudes except for danger-related beliefs. That is, those who viewed the non-categorical poster reported an increased perception that people with mental health problems are dangerous.

### Discussion

Our largely null findings may suggest the equivalence of these posters on stigma and help-seeking intentions but may also reflect the brevity of the intervention. Our findings concerning danger beliefs may reflect a Type I error, the complexities of stigma models, or the

**Funding:** The author(s) received no specific funding for this work.

**Competing interests:** The authors have declared that no competing interests exist.

adverse effects of increased perceived contact. Further research is needed to test the effects of differing mental health paradigms on stigma and help-seeking intentions over a longer duration.

## 1. Introduction

Mental health stigma can be defined as a combination of a lack of knowledge, and prejudice and discrimination against those with mental health difficulties [1]. Cognitive models of mental health stigma have identified key attitudes as predictive of discriminatory behaviours. Specifically, increased stigma is associated with perceiving those with mental health problems are dangerous and personally responsible for their symptoms [2–4].

The impacts of mental health stigma on people are significant and wide-reaching, with personal, social and economic repercussions. For example, mental health stigma can impede finances by reducing employability [5], it can worsen symptoms due to delayed help-seeking [6], and it can increase isolation due to fears and experiences of judgement and rejection [2, 7]. Anti-stigma campaigns have therefore tried to reduce public stigma towards people with mental health problems. These campaigns make use of visual representations to portray messages. Images are an effective way of communicating mental health-related information as they are thought to encourage elaborative thinking [8]. One of the largest anti-stigma campaigns is the UK's "Time to Change" (TTC) programme [9]. TTC was launched in 2009 and evaluations have generally shown incremental improvements in reducing public stigma around mental health with each year of its existence [10–12]. However, interim analysis assessing the relationship between campaign awareness and changes in specific aspects of stigma demonstrate that neither tolerance [10] or prejudice [11] was associated with TTC awareness.

The shortcomings of the TTC may be explained by its inherent alignment with a psychiatric understanding of mental health. That is, the content of TTC campaigns adhered to the 'medical model' i.e. that mental health difficulties can be understood and categorised using and symptom thresholds [13]. TTC mirrored this approach in their frequent mention of clinical diagnoses and the delineation of those with and without mental health problems in their primary tagline of "1 in 4 people will experience a mental health problem in their lifetime". Such prevalence statistics describe the number of people with versus without a particular symptom or characteristic [14]–therein implying a categorical perspective of mental health with one person having a mental illness, while the other three do not. The emphasis on categorising people into groups differentiated by having or not having mental health problems creates a sense of "otherness", which is associated with increased stigma [15, 16]. Thus, the use of categorisation in TTC campaign messaging, although intended to emphasise the commonality of mental health problems, may inadvertently have undermined any potential de-stigmatising effects.

In a recent review, the opposing perspective of a non-categorical approach (also referred to as the continuum approach) was shown to generally be associated with a reduction in mental health stigma [17]. But while this approach may be superior in reducing stigma, it may have some unintended negative consequences. There is some limited literature suggesting that over-normalising mental health problems (i.e. removing any notion of "otherness") can adversely impact help-seeking [18, 19]. The proposed explanation is that mental distress becomes accepted as a 'normal' human experience that does not require any support or intervention. Delayed help-seeking as a consequence of over-normalising can have life or death

consequences [20]. Without mental health support, symptoms are likely to be prolonged and get worse [21].

The review by Peter et al. [17] brought together the findings of eight intervention studies, three of which were with members of the public as participants. However, Peter et al. [17] highlights these studies are limited in that they did not manipulate allocation to the intervention message and therefore cannot provide any causal evidence of their impact of non-categorical beliefs on stigma. To our knowledge, there is currently no experimental test of the impact of a categorical versus non-categorical anti-stigma campaign that assesses its impact on both stigma outcomes and help-seeking intentions.

The TTC came to an end in 2021 and mental health stigma is still prevalent, leaving space for a new and improved mental health public stigma campaign. The current study aims to compare the categorical verses non-categorical approaches to anti-stigma messaging with respect to their effects on stigma and help-seeking intentions.

### 1.1 Research hypotheses

The present study will aim to test the following hypotheses:

1. After controlling for familiarity, country of residence, and baseline scores, mental health stigma attitudes will be less negative for those who attend to the non-categorical anti-stigma poster.

2. After controlling for familiarity, country of residence, and baseline scores, help-seeking intentions will be greater for those attending to the categorical anti-stigma poster.

## 2. Materials and methods

### 2.1 Design

We conducted an online survey with an embedded pre-post Experimental design with two independent groups: categorical versus non-categorical anti-stigma poster. Participants were randomly assigned to view one of these posters using the randomisation function within Qualtrics, aiming for a 1:1 group allocation ratio. This study is reported using the CONSORT-SPI 2018 Extension guidelines [22].

### 2.2 Participants

To be eligible to participate in this survey, persons had to self-identify as aged 16 or over, and able to read and write in English. Participants were not limited to a particular country–we therefore controlled for country of residence in our analysis. We produced an advert for the study and posted this across social media channels and online forums, as well as encouraging snowball recruitment.

### 2.3 Anti-stigma poster

Participants were randomised to view one of two anti-stigma posters: either the categorical poster or the non-categorical poster (Fig 1).

The categorical poster aligns with the medical model of mental health whereby good and poor mental health can be clearly delineated with cut-offs that describe the person as either having or not having a diagnosable mental health problem [23, 24]–this poster is akin to the sentiment of the Time to Change campaign [25]. The non-categorical poster moves away from cut-offs and/or diagnostic labels and instead emphasises that we all have mental health and that this is fluctuating and changeable–this poster reflects the message of the "Only Us"

**Categorical poster:**

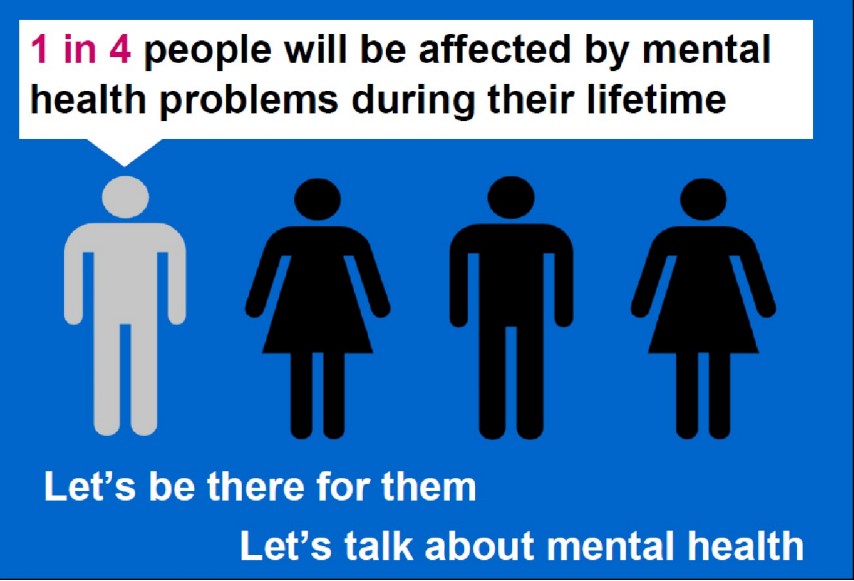

**Non-categorical poster:**

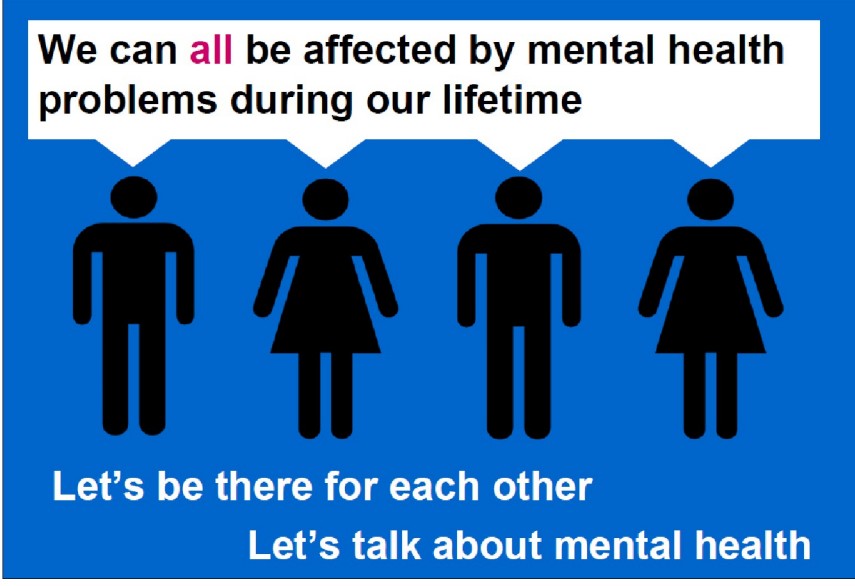

**Fig 1. Mental health stigma posters.**

campaign [26]. An explanation of how we communicated these differing perspectives on mental health is outlined in Table 1.

## 2.4. Measures

**2.4.1 Attribution Questionnaire.** Mental health stigma attitudes were assessed using the Attribution Questionnaire (AQ) [3]. Other versions of the AQ ask participants about their mental health attitudes in relation to a specific patient vignette [27]. However, we have used

**Table 1. Details of how the anti-stigma posters differed between experimental conditions.**

| Poster Element | Categorical | Non-categorical |
|---|---|---|
| People graphic | One of the four people are in a different colour suggesting they have a mental health problem, whereas the other three do not. | All of the people are the same colouring suggesting that no one is different. |
| Speech bubble | The "1 in 4" suggests that people can either have or do not have a mental health problem i.e. there are cut-offs. | The "all" represents the idea that poor mental health can be experienced by everyone and is not necessarily stable. |
| Tagline | The use of "them" is intended to reinforce the idea of separation between those who do and those who do not have a mental health problem. | There is no "other" identified here and instead poor mental health can be ubiquitous. |

the non-specific version of the AQ that asks participants about their attitudes to those with mental health problems more generally [3]. This version of the AQ has 20 items and can be divided into seven subscales that each reflect a different attitude: (1) personal responsibility: perception of how much control the person has over their mental health; (2) pity: how much sympathy they have towards people with mental health problems; (3) helping behaviour: willingness to help people with mental health problems; (4) anger: how angry they feel towards people with mental health problems; (5) dangerousness: the perception of how much threat people with mental health problems pose to them; (6) fear: how afraid they are of people with mental health problems; (7) avoidance: how much they want to avoid people with mental health problems. The seven-factor structure was found to have statistically significant good model fit ($p < .001$).

**2.4.2 Inventory of Attitudes towards Seeking Mental Health Services (IASMHS).** The Inventory of Attitudes towards Seeking Mental Health Services (IASMHS) [28] is a 24 item questionnaire measuring the extent to which persons would seek help if they were to experience a mental health problem. The IASMHS is comprised of three sub-scales: (1) psychological openness: how open a person would be to discussing their emotions; (2) help-seeking propensity: willingness to seek help generally, and (3) indifference to stigma: whether fear of stigma will prevent help-seeking. Participants rated their agreement with statements using a Likert scale from 0 (disagree) to 4 (agree). The subscales can also be totalled to give an overall score of attitudes towards seeking mental health help. All of the subscales and scale total were found to have good reliability ($\alpha s \geq .76$) in both the original study [28], and in a subsequent re-evaluation of the IASMHS [29].

**2.4.3 Familiarity.** The posters were based on existing mental health campaign posters. To enable us to control for familiarity with the posters, we included a one-item visual analogue scale. Participants rated their familiarity with the poster from 0 –"totally unfamiliar, I have never seen it before", to 100 –"totally familiar, I have definitely seen it before".

**2.4.4 Attention checks.** We ensured that participants had attended to the poster by asking them three multiple choice questions testing their knowledge of the poster content. We asked participants: (1) According to the poster, how many people are affected by mental health problems during their lifetime?; (2) Select the word that is missing from this caption found on the image above: "Let's be there for [blank]"; and, (3) What does this image want you to do to help reduce mental health stigma?.

## 2.5 Procedure

After providing consent and completing demographic questions, participants were asked to complete the aforementioned measures of mental health stigma attitudes and help-seeking

intentions (T0 assessment). Participants were then randomised to view one of the anti-stigma posters and complete the attention check questions. Participants were randomised using the 'randomise' function within Qualtrics with an allocation ratio of 1:1. Participants were then asked to complete the same mental health stigma attitudes and help-seeking intentions questionnaires (T1 assessment). Participants were finally presented with a debrief statement and given the opportunity to enter a prize draw to win one of five £20 prizes.

### 2.6 Analysis plan

Participants who did not correctly answer the attention check questions were excluded from the analysis. To test the research hypotheses, we conducted a one-way MANCOVA where familiarity with the poster, a dummy variable for country of residence (UK versus rest of the world), and T0 AQ subscales [3] and IASMHS subscales and scale total [28] scores were entered as covariates. The type of anti-stigma poster (categorical versus non-categorical) was entered as the independent variable, and the T1 AQ subscales [3] and IASMHS subscales and scale total [28] were entered as dependent variables. We report the Pillai's Trace effect size.

### 2.7 Ethics

This study received ethical approval as part of a larger online mental health survey from the University of Sussex Research Ethics Committee (reference: ER/CH283/8). Participants provided online informed consent by completing a tick box form.

## 3 Results

### 3.1 Sample characteristics

A total of 1,570 participants completed the consent statement and provided some demographic information. After removing those who did not view the anti-stigma poster or correctly complete the attention check, this left a final sample of 1,046 participants (Fig 2).

Our sample was largely female, White British, living in the United Kingdom, employed, and with an academic qualification (see Table 2).

### 3.2 Overall model

After controlling for covariates, there was no significant difference between those participants who viewed the categorical versus non-categorical stigma poster on a composite of all the stigma and help-seeking scales and subscales ($F(12, 932) = 0.86$, $p = .59$; $V = .01$).

### 3.3 Mental health stigma

After controlling for covariates, there was no significant difference between the categorical and non-categorical poster groups in terms of the personal responsibility ($F(1, 943) = 0.001$, $p = .97$), pity ($F(1, 943) = 0.08$, $p = .78$), anger ($F(1, 943) = 0.70$, $p = .40$), helping behaviour ($F(1, 943) = 0.03$, $p = .87$), fear ($F(1, 943) = 0.30$, $p = .58$), or avoidance ($F(1, 943) = <0.001$, $p = 1.00$) subscales. There was a significant group effect on danger-related attitudes ($F(1, 943) = 5.15$, $p = .02$), whereby those who viewed the non-categorical poster reported a great belief that people with mental health problems were dangerous than those who viewed the categorical poster.

### 3.4 Help-seeking intentions

After controlling for covariates, there was no significant difference between the participants who viewed the categorical versus the non-categorical poster on the IASMHS scale total ($F(1$,

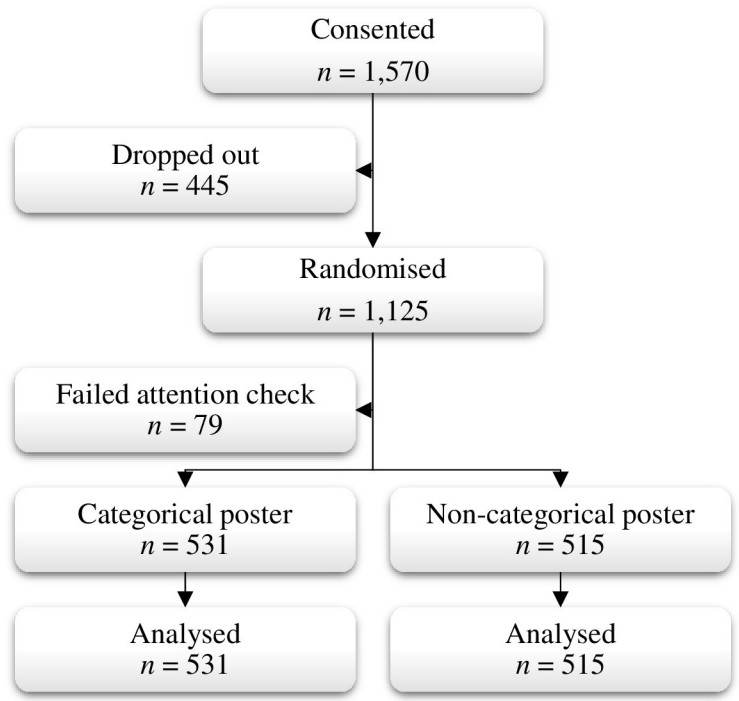

**Fig 2. CONSORT diagram.**

943) = 0.01, $p$ = .92), or psychological openness ($F(1, 943)$ = 1.12, $p$ = .29), help-seeking propensity ($F(1, 943)$ = 0.20, $p$ = .66), and indifference to stigma ($F(1, 943)$ = 0.04, $p$ = .84) subscales.

## 4 Discussion

The aim of our study was to compare the effects of two mental health anti-stigma posters on self-reported stigma and help-seeking intentions for mental health difficulties. The two posters adopted differing approaches to the conceptualisation of mental health problems: one adopting a categorical perspective and the other a non-categorical perspective. We found that there was no difference between the two groups in terms of a composite measure of stigma and help-seeking intentions derived via a MANOVA, help-seeking intentions alone, or most of the mental health stigma attitudes assessed here. The only construct where the two groups differed were danger related beliefs. Contrary to our hypotheses, viewing the non-categorical poster was associated with increased endorsement that people with mental health problems are dangerous.

Largely we found there were no differences between our two groups in terms of stigma or help-seeking intentions. The null findings may suggest the posters have equivalent effects but may also reflect the brevity of our intervention. Participants only viewed the campaign poster for a short amount of time–just long enough to answer the attention check questions. Similarly, another brief anti-stigma campaign was found to have limited efficacy, producing improvements only on knowledge-related outcomes [30]. To produce any changes on attitudinal (e.g., beliefs about people with mental health problems) or behavioural (e.g., help-seeking intentions) outcomes is likely to require a longer-term intervention.

One variable that we did find a significant between-group difference on were danger-related beliefs. The literature largely criticises anti-stigma campaigns for promoting

**Table 2. Descriptive statistics of the sample characteristics and research data.**

| | | Categorical | | Non-categorical | | All | |
|---|---|---|---|---|---|---|---|
| | *n* | *M(SD)* or *n(%)* | *n* | *M(SD)* or *n(%)* | *n* | *M(SD)* or *n(%)* | |
| **Sample characteristics** | | | | | | | |
| Age *M(SD)* | 531 | 32.30(12.62) | 515 | 32.76(13.10) | 1046 | 32.53(12.86) | |
| Gender *n(%)* | | | | | | | |
| *Male* | | 102(19.2) | | 111(21.6) | | 213(20.4) | |
| *Female* | | 419(78.9) | | 393(76.3) | | 812(77.6) | |
| *Other* | | 8(1.5) | | 9(1.8) | | 17(1.6) | |
| *Prefer not to say* | | 2(0.4) | | 2(0.4) | | 4(0.4) | |
| Ethnicity *n(%)* | 531 | | 515 | | 1046 | | |
| *White British or White other* | | 464(87.4) | | 453(88.0) | | 917(87.6) | |
| *Black British or Black other* | | 7(1.3) | | 7(1.4) | | 14(1.4) | |
| *Asian British or Asian other* | | 21(3.9) | | 14(2.8) | | 35(3.4) | |
| *Indian British or Indian other* | | 10(1.9) | | 14(2.7) | | 24(2.3) | |
| *Mixed ethnicity* | | 21(4.0) | | 22(4.3) | | 43(4.1) | |
| *Prefer not to say* | | 8(1.5) | | 5(1.0) | | 13(1.2) | |
| Country of birth *n(%)* | 531 | | 515 | | 1046 | | |
| *England* | | 313(58.9) | | 311(60.4) | | 624(59.7) | |
| *Scotland* | | 32(6.0) | | 27(5.2) | | 59(5.6) | |
| *Wales* | | 7(1.3) | | 12(2.3) | | 19(1.8) | |
| *Northern Ireland* | | 7(1.3) | | 3(0.6) | | 10(1.0) | |
| *Republic of Ireland* | | 9(1.7) | | 6(1.2) | | 15(1.4) | |
| *Elsewhere* | | 163(30.7) | | 156(30.3) | | 319(30.5) | |
| First language *n(%)* | 531 | | 515 | | 1046 | | |
| *English* | | 447(84.2) | | 446(86.6) | | 893(85.4) | |
| *Not English* | | 59(11.1) | | 47(9.1) | | 106(10.1) | |
| *Bilingual from birth* | | 25(4.7) | | 21(4.1) | | 46(4.4) | |
| *Prefer not to say* | | 0(0) | | 1(0.2) | | 1(0.1) | |
| Marital status *n(%)* | 531 | | 515 | | 1046 | | |
| *Single* | | 209(39.4) | | 212(41.2) | | 421(40.2) | |
| *Married, civil partnership, cohabiting or in a relationship* | | 288(54.3) | | 271(52.6) | | 559(53.4) | |
| *Divorced or separated* | | 25(4.7) | | 25(4.9) | | 50(4.8) | |
| *Widowed* | | 5(0.9) | | 2(0.4) | | 7(0.7) | |
| *Prefer not to say* | | 4(0.8) | | 5(1.0) | | 9(0.9) | |
| Sexual orientation *n(%)* | 531 | | 515 | | 1046 | | |
| *Heterosexual or straight* | | 422(79.5) | | 389(75.5) | | 811(77.5) | |
| *Homosexual or gay* | | 22(4.2) | | 27(5.2) | | 49(4.7) | |
| *Bisexual* | | 59(11.1) | | 55(10.7) | | 114(10.9) | |
| *Other* | | 10(1.9) | | 16(3.1) | | 26(2.5) | |
| *Unsure* | | 11(2.1) | | 18(3.5) | | 29(2.8) | |
| *Prefer not to say* | | 7(1.3) | | 10(1.9) | | 17(1.6) | |
| Employment status *n(%)* | 531 | | 515 | | 1046 | | |
| *Employed (paid)* | | 270(50.8) | | 255(49.5) | | 525(50.2) | |
| *Employed (voluntary)* | | 7(1.3) | | 6(1.2) | | 13(1.2) | |
| *Student* | | 193(36.3) | | 191(37.1) | | 384(36.7) | |
| *Homemaker* | | 16(3.0) | | 25(4.9) | | 41(3.9) | |
| *Unemployed* | | 36(6.8) | | 29(5.6) | | 65(6.2) | |
| *Prefer not to say* | | 9(1.7) | | 9(1.7) | | 18(1.7) | |

*(Continued)*

**Table 2.** (Continued)

|  |  | | Categorical | | Non-categorical | | All |
| --- | --- | --- | --- | --- | --- | --- | --- |
|  |  | *n* | *M(SD)* or *n*(%) | *n* | *M(SD)* or *n*(%) | *n* | *M(SD)* or *n*(%) |
|  | Highest qualification *n*(%) | 531 |  | 515 |  | 1046 |  |
|  | *No qualification* |  | 15(2.8) |  | 10(1.9) |  | 25(2.4) |
|  | *GCSE or equivalent* |  | 28(5.3) |  | 35(6.8) |  | 63(6.0) |
|  | *A level or equivalent* |  | 128(24.1) |  | 139(27.0) |  | 267(25.5) |
|  | *Undergraduate degree or equivalent* |  | 198(37.3) |  | 179(34.8) |  | 377(36.0) |
|  | *Postgraduate degree or equivalent* |  | 124(23.4) |  | 126(24.5) |  | 250(23.9) |
|  | *Doctoral degree or equivalent* |  | 26(4.9) |  | 20(3.9) |  | 46(4.4) |
|  | *Prefer not to say* |  | 12(2.3) |  | 6(1.2) |  | 18(1.7) |
| **AQ** |  |  |  |  |  |  |  |
|  | Personal responsibility | 494 | 2.06(1.26) | 465 | 2.13(1.36) | 959 | 2.09(1.31) |
|  | Pity | 494 | 1.79(1.02) | 465 | 1.81(1.11) | 959 | 1.80(1.06) |
|  | Helping behaviour | 494 | 2.10(1.29) | 465 | 2.12(1.33) | 959 | 2.11(1.31) |
|  | Anger | 494 | 3.32(1.26) | 465 | 3.34(1.33) | 959 | 3.33(1.29) |
|  | Dangerousness | 494 | 7.13(1.68) | 465 | 7.27(1.50) | 959 | 7.20(1.60) |
|  | Fear | 494 | 3.58(1.12) | 465 | 3.59(1.16) | 959 | 3.58(1.14) |
|  | Avoidance | 494 | 5.63(1.32) | 465 | 5.62(1.07) | 959 | 5.63(1.20) |
| **IASMHS** |  |  |  |  |  |  |  |
|  | Total | 494 | 1.82(0.45) | 465 | 1.82(0.40) | 959 | 1.82(0.43) |
|  | Psychological openness | 494 | 1.12(0.80) | 465 | 1.09(0.76) | 959 | 1.11(0.78) |
|  | Help-seeking propensity | 494 | 3.01(0.86) | 465 | 3.02(0.80) | 959 | 3.02(0.83) |
|  | Indifference to stigma | 494 | 1.32(0.86) | 465 | 1.34(0.83) | 959 | 1.33(0.85) |

*Note*: *M* = mean; *SD* = standard deviation; AQ = Attribution Questionnaire (Corrigan et al., 2002); IASMHS = Inventory of Attitudes towards Seeking Mental Health Support [28].

"otherness" by portraying those with mental health difficulties as different [31]. Instead, those that adopt a recovery-orientated approach and advocate inclusiveness are thought to be the most effective and acceptable campaigns [32]. Our findings perhaps contradict this message, as we found danger-related attitudes were increased in those that viewed the non-categorical poster. Given all other results were non-significant, we believe this result is likely to be artefact, reflecting a Type I error. Potentially though, our results might be explained by the complexities and nuances of how mental health paradigms, stigma, and help-seeking are defined and the relationships between these constructs. For example, while believing people with mental health problems are dangerous and blameworthy are both stigmatising attitudes, their effects on help-seeking are distinct. The former of these attitudes (danger beliefs) is associated with increased help-seeking while the latter (personal responsibility beliefs) is associated with reduced help-seeking [33]. It is therefore possible that mental health stigma campaigns may produce effects on individual stigmatising attitudes. Assuming this is correct, the present non-categorical poster may have had a specific impact on fear-related beliefs by enhancing the perceived proximity of those with mental health problems. That is, people with mental health problems are not "other" but are instead part of their in-group. Being 'close' to someone with mental health problems may therefore specifically increase the perceived likelihood of threat resulting in increased fear [34]. Our results here may reflect the adverse consequences of contact on specific mental health related attitudes.

### 4.1 Limitations

A limitation of our experimental manipulation is the brevity of participants' exposure to the anti-stigma campaigns. Participants were required to attend to the poster only for long enough to complete the attention checks correctly. Our assessment of stigma is also limited in that the version of the Attribution Questionnaire (AQ) [3] used here does not consider the heterogeneity of mental health problems. Self-reported stigma varies in relation to the presence and nature of a clinical diagnosis [35], therefore calling into question the utility of non-specific measures of mental health stigma like that used here. For example, studies suggest that a 'hierarchy' of stigma exists for psychiatric conditions [36], with certain 'labels,' such as schizophrenia, generally more feared and stigmatized than others such as depression [37]. Further issues related to our method of measurement is that the wording of the AQ may have tainted our experimental manipulation–especially for the non-categorical poster arm. The language used in the AQ may have subtly communicated an alignment with the categorical approach i.e. referring to people with mental health problems, suggesting an "otherness". Participants completing this questionnaire in the non-categorical arm will have viewed potentially contradictory ideas and the impact of this on their responses cannot be fully determined. Addressing this in future research studies will be a challenge for researchers who must consider ways of measuring endorsement of competing ideologies using neutral language. Implications:

Our null findings mean we cannot offer any suggestion as to which approach, categorical or non-categorical, is most effective at reducing mental health stigma and encouraging help-seeking when needed. Previous literature suggests a superiority of the non-categorical (continuum) approach, but these intervention studies are limited by their use of cross-sectional rather than experimental designs [17]. We therefore do not yet know if there is an approach to anti-stigma campaigns that can surpass the gains achieved by the Time to Change approach. As this question remains unanswered, we assert that an experimental test assessing the effectiveness of opposing mental health stigma campaigns is still very much needed. However, such a test needs to assess the effectiveness of these campaigns over a prolonged period of time. We also recommend that further research in this area should take into consideration the heterogeneous nature of mental health difficulties–in terms of both the content of anti-stigma campaigns and measures to assess their effectiveness. On reflection, it is unlikely that such a simple intervention is likely to bring about significant and sustained changes in a construct as complex and multifaceted as stigma. Negative depictions of mental health have been prevalent for a long period of time [38] and are embedded in mainstream media (e.g. newspapers and magazines [39] and television [40]). Ultimately, it is more probable that a successful stigma campaign will require a multipronged approach that offers a degree of personalisation in terms of the type of mental health problem and the context.

## 5 Conclusion

The aim of this study was to compare the effectiveness of two posters that adhere to two different theoretical perspectives of mental health on mental health stigma and help-seeking intentions. Contrary to our hypotheses, we found no significant difference between the posters on all variables bar danger-related beliefs; however, this result is likely a Type I error. To establish which approach is best for future mental health stigma campaigns, further experimental studies are needed that evaluate campaigns that involve longer term exposure to the messaging.

## Supporting information

**S1 File.**
(SAV)

## Acknowledgments

Thank you to Dr Sarah Fielding-Smith, Ciara Gavurin, Hannah Fell, Hannah Newcombe, and Lucy Mainord for their support in recruiting participants for this survey. Thank you to the participants for taking the time to participant in this study.

## Author Contributions

**Conceptualization:** Cassie M. Hazell, Clio Berry.

**Data curation:** Cassie M. Hazell, Clio Berry.

**Formal analysis:** Cassie M. Hazell.

**Investigation:** Cassie M. Hazell.

**Methodology:** Cassie M. Hazell, Clio Berry.

**Project administration:** Cassie M. Hazell.

**Writing – original draft:** Cassie M. Hazell.

**Writing – review & editing:** Cassie M. Hazell, Alison Fixsen, Clio Berry.

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
