## [Decision Letter · Decision Letter 0]

2 May 2022

PONE-D-22-07331Is it time to change the approach of mental health stigma campaigns? An experimental investigation of the effect of campaign wording on stigma and help-seeking intentions.PLOS ONE

Dear Dr. Hazell,

Thank you for submitting your manuscript to PLOS ONE. After careful consideration, we feel that it has merit but does not fully meet PLOS ONE’s publication criteria as it currently stands. Therefore, we invite you to submit a revised version of the manuscript that addresses the points raised during the review process.

ACADEMIC EDITOR: Please read and respond to the reviewers' comments, and revise your manuscript.According to the journal's guideline, the authors need to  conform to appropriate reporting guidelines (e.g. CONSORT for this study): https://www.equator-network. Please also consider providing the relevant checklist as a supporting information.==============================

We look forward to receiving your revised manuscript.

Kind regards,

Naoki Yoshinaga

Academic Editor

PLOS ONE

Journal Requirements:

Additional Editor Comments:

According to the journal's guideline, the authors need to conform to appropriate reporting guidelines (e.g. CONSORT for this study): https://www.equator-network.

Please also consider providing the relevant checklist as a supporting information.

Reviewers' comments:

Reviewer's Responses to Questions

**Comments to the Author**

1. Is the manuscript technically sound, and do the data support the conclusions?

Reviewer #1: Yes

Reviewer #2: Yes

2. Has the statistical analysis been performed appropriately and rigorously? 

Reviewer #1: Yes

Reviewer #2: Yes

3. Have the authors made all data underlying the findings in their manuscript fully available?

Reviewer #1: No

Reviewer #2: No

4. Is the manuscript presented in an intelligible fashion and written in standard English?

Reviewer #1: Yes

Reviewer #2: Yes

5. Review Comments to the Author

Reviewer #1: Overall this is a well written paper on an important topic. The language is accessible and clear throughout. I’ve made a couple suggestions for how the authors may like to develop the paper, but these really are suggestions rather than requests. Thank you, it was a pleasure to read your paper. I recommend it’s publication.

Abstract:

The abstract is provides a clear description of the need for research, the methods used, the results found, and possible interpretations. At the moment the abstract is long and might benefit from being condensed to key points.

Introduction:

Excellent description of the need to challenge mental health-related stigma, current dominant strategies (e.g. TTC campaign), and the potential for advancing alternative approaches.

The paper, and subsequent research hypotheses, would benefit from engaging with theories of health communication to explain the theoretical model (e.g. Theory of planned behaviour; health belief model; social representations theory etc) by which visual communication is considered to influence individual attitudes.

Clear description of research hypotheses

Methods:

Clear description of research design, participants, and posters. Measures are well described and relevant. Clear account of analysis plan; relevant choice of MANCOVA for addressing increased Type I error

Results:

Clear descriptive statistics of the sample characteristics and research data; and researchers effectively control for covariates.

Discussion:

Clear interpretation of the results.

Slight issues with wording on line 230: researchers states “The null findings may suggest the posters have equivalent effects, but are likely a result of the brevity of our intervention” as this statement was not directly evidenced, it might be more approach to say “may reflect the brevity” rather than “likely”.

Interesting and balanced discussion of the ‘fear effect’. These is some research e.g. Thibodeau & Peterson (2018) that continuum-belief interventions to increase participants experiences of anxiety and threat. There is also evidence e.g. Foster, 2001; Walsh & Foster, 2020 that perceiving the Other as relevant to the Self is experienced as threatening for one’s social identity (and empowered position in the moral order).

Implications:

To support the statement on line 275 - 277: “On reflection, it is unlikely that such a simple intervention is likely to bring about significant and sustained changes in a construct as complex and multifaceted as stigma” It might be worth also highlighting that representations of mental health problems are historically tenacious (Jodelet, 1991) and very much embedded in institutions (e.g. the media) (Rose, 1998).

Reviewer #2: Introduction

The summary of the impact of Time to Change is based on results published part way into the campaign, which finished in 2021. For the results published towards the end ie after more changes had accrued, see

Henderson C, Potts L, Robinson EJ. Mental illness stigma after a decade of Time to Change England: inequalities as targets for further improvement. European Journal of Public Health, 30 (3): 526–532, https://doi.org/10.1093/eurpub/ckaa013, 2020.

The discussion about a categorical vs noncategorial approach needs more work. First, emphasising the prevalence of mental health problems is a way to normalise them and raise people’s awareness of common mental disorder, the symptoms of which are at the severe end of experiences that everyone has at some point, therefore the use of a prevalence message is not purely in opposition to a noncategorical one. Second, the body of work by Georg Schomerus and others on the continuum model is entirely neglected; instead the hypotheses are presented as though no one had tested them before.

Methods

As this is an online RCT I recommend the use of the online RCT extension of the Consort reporting guidance, including a consort flow diagram.

The categorical poster description of diagnosis/problem appears to assume that these are synonymous in people’s minds. This is not the case- many people think of nondiagnostic issues as mental health problems. This is something that increased over Time to Change, the evaluation of which included assessment of whether people think of grief and stress as mental health problems.

The AQ ask about people with mental health problems, implying a categorical attitude. Is this a problem for measurement of the impact of the noncategorical poster?

Discussion

Given the evidence cited for the lack of impact of short term campaigns on the outcomes of interest it is not clear what the rationale for the current study was.

The implications section again ignores the existing body of work on the continuum model. How does the current study build on the work of Schomerus et al? Without such contextualisation and given the likely effect of a high prevalence message on reducing othering it is hard to see what this study adds.

6. PLOS authors have the option to publish the peer review history of their article (what does this mean?). If published, this will include your full peer review and any attached files.

Reviewer #1: **Yes: **Dr Daniel Walsh

Reviewer #2: **Yes: **Dr Claire Henderson

---

## [Author Response · Author response to Decision Letter 0]

28 Jun 2022

We have attached our response to the reviewers as a cover letter. The unformatted version of the cover letter is copied and pasted here also:

Manuscript Title: Is it time to change the approach of mental health stigma campaigns? An experimental investigation of the effect of campaign wording on stigma and help-seeking intentions.

We thank the reviewers for their thoughtful and important comments. We are pleased to have the opportunity to revise our paper in light of their feedback. We have made changes to our manuscript using tracked changes. Our response letter is formatted with the reviewers’ comments given in bold, our response in normal typeface, and where needed, extracts from the paper italicised.

EDITOR COMMENTS:

We have reviewed the guidance and have updated the manuscript to adhere to this guidance.

We have included a statement to the ethics section of the method describing how consent was taken:

Participants provided online informed consent by completing a tick box form. 

3. We note that you have indicated that data from this study are available upon request. PLOS only allows data to be available upon request if there are legal or ethical restrictions on sharing data publicly. For more information on unacceptable data access restrictions, please see http://journals.plos.org/plosone/s/data-availability#loc-unacceptable-data-access-restrictions. In your revised cover letter, please address the following prompts:

a. If there are ethical or legal restrictions on sharing a de-identified data set, please explain them in detail (e.g., data contain potentially sensitive information, data are owned by a third-party organization, etc.) and who has imposed them (e.g., an ethics committee). Please also provide contact information for a data access committee, ethics committee, or other institutional body to which data requests may be sent.

b. If there are no restrictions, please upload the minimal anonymized data set necessary to replicate your study findings as either Supporting Information files or to a stable, public repository and provide us with the relevant URLs, DOIs, or accession numbers. For a list of acceptable repositories, please see http://journals.plos.org/plosone/s/data-availability#loc-recommended-repositories.

We are happy to share the minimal anonymised data set from this study. Thank you for updating the data availability statement as required. 

The corresponding author has an ORCID iD that can be attached to this submission: 0000-0001-58689902; however, I have been unable to do this when submitting and resubmitting the paper. I receive an error message each time I try to connect the ORCID iD. Any assistance you can provide in fixing this would be appreciated. 

5. According to the journal's guideline, the authors need to conform to appropriate reporting guidelines (e.g. CONSORT for this study): https://www.equator-network.

This paper has been written in line with the CONSORT-SPI 2018 Extension guidelines. This has been included in the methods section. 

REVIEWER 1 COMMENTS:

6. Overall this is a well written paper on an important topic. The language is accessible and clear throughout. I’ve made a couple suggestions for how the authors may like to develop the paper, but these really are suggestions rather than requests. Thank you, it was a pleasure to read your paper. I recommend it’s publication.

We would like to thank the reviewer for their kind comments about our paper. 

7. Abstract: The abstract is provides a clear description of the need for research, the methods used, the results found, and possible interpretations. At the moment the abstract is long and might benefit from being condensed to key points.

We have now revised the abstract to reduce its length. The abstract is now below 250 words. 

8. Introduction: Excellent description of the need to challenge mental health-related stigma, current dominant strategies (e.g. TTC campaign), and the potential for advancing alternative approaches.

We would like to thank the reviewer for their kind comments about our paper. 

9. Introduction: The paper, and subsequent research hypotheses, would benefit from engaging with theories of health communication to explain the theoretical model (e.g. Theory of planned behaviour; health belief model; social representations theory etc) by which visual communication is considered to influence individual attitudes.

We have included a sentence to the introduction to give a theoretical basis for the use of poster in anti-stigma campaigns:

Anti-stigma campaigns have therefore tried to reduce public stigma towards people with mental health problems. These campaigns make use of visual representations to portray messages. Images are an effective way of communicating mental health-related information as they are thought to encourage elaborative thinking (Lazard et al., 2016).

10. Methods: Clear description of research design, participants, and posters. Measures are well described and relevant. Clear account of analysis plan; relevant choice of MANCOVA for addressing increased Type I error.

We would like to thank the reviewer for their kind comments about our paper. 

11. Results: Clear descriptive statistics of the sample characteristics and research data; and researchers effectively control for covariates.

We would like to thank the reviewer for their kind comments about our paper. 

12. Discussion: Slight issues with wording on line 230: researchers states “The null findings may suggest the posters have equivalent effects, but are likely a result of the brevity of our intervention” as this statement was not directly evidenced, it might be more approach to say “may reflect the brevity” rather than “likely”. 

We have amended the wording as follows:

The null findings may suggest the posters have equivalent effects, but may also reflect the brevity of our intervention.

13. Discussion: Interesting and balanced discussion of the ‘fear effect’. These is some research e.g. Thibodeau & Peterson (2018) that continuum-belief interventions to increase participants experiences of anxiety and threat. There is also evidence e.g. Foster, 2001; Walsh & Foster, 2020 that perceiving the Other as relevant to the Self is experienced as threatening for one’s social identity (and empowered position in the moral order).

We have amended this section of the discussion to include some of the literature described by the reviewer: 

For example, while believing people with mental health problems are dangerous and blameworthy are both stigmatising attitudes, their effects on help-seeking are distinct. The former of these attitudes (danger beliefs) is associated with increased help-seeking while the latter (personal responsibility beliefs) is associated with reduced help-seeking (Mojtabai, 2010). It is therefore possible that mental health stigma campaigns may produce effects on individual stigmatising attitudes. Assuming this is correct, the present non-categorical poster may have had a specific impact on fear-related beliefs by enhancing the perceived proximity of those with mental health problems. That is, people with mental health problems are not “other” but are instead part of their in-group. Being ‘close’ to someone with mental health problems may therefore specifically increase the perceived likelihood of threat resulting in increased fear (Thibodeau & Peterson, 2018). Our results here may reflect the adverse consequences of contact on specific mental health related attitudes.

14. Implications: To support the statement on line 275 - 277: “On reflection, it is unlikely that such a simple intervention is likely to bring about significant and sustained changes in a construct as complex and multifaceted as stigma” It might be worth also highlighting that representations of mental health problems are historically tenacious (Jodelet, 1991) and very much embedded in institutions (e.g. the media) (Rose, 1998).

We have added a sentence to the discussion to reflect on the longstanding and pervasive nature of mental health stigma: 

On reflection, it is unlikely that such a simple intervention is likely to bring about significant and sustained changes in a construct as complex and multifaceted as stigma. Negative depictions of mental health have been prevalent for a long period of time (Schomerus & Angermeye, 2017) and are embedded both implicitly and explicitly in mainstream media (e.g. newspapers and magazines (Nawková et al., 2012) and television (Henderson, 2018)).

REVIEWER 2 COMMENTS:

15. Introduction: The summary of the impact of Time to Change is based on results published part way into the campaign, which finished in 2021. For the results published towards the end ie after more changes had accrued, see Henderson C, Potts L, Robinson EJ. Mental illness stigma after a decade of Time to Change England: inequalities as targets for further improvement. European Journal of Public Health, 30 (3): 526–532, https://doi.org/10.1093/eurpub/ckaa013, 2020.

Thank you to the reviewer for highlighting this reference. We have added it to our introduction:

TTC was launched in 2009 and evaluations have generally shown incremental improvements in reducing public stigma around mental health with each year of its existence (Evans-Lacko et al., 2014; Henderson et al., 2020; Sampogna et al., 2017).

16. Introduction: The discussion about a categorical vs noncategorial approach needs more work. First, emphasising the prevalence of mental health problems is a way to normalise them and raise people’s awareness of common mental disorder, the symptoms of which are at the severe end of experiences that everyone has at some point, therefore the use of a prevalence message is not purely in opposition to a noncategorical one. 

We appreciate that prevalence statistics and the categorical approach to mental health are not necessarily interchangeable. We have included further information about how we do believe in this instance the “1 in 4” message does communicate a categorical understanding of mental health problems: 

TTC mirrored this approach in their frequent mention of clinical diagnoses and the delineation of those with and without mental health problems in their primary tagline of “1 in 4 people will experience a mental health problem in their lifetime”. Such prevalence statistics describe the number of people with versus without a particular symptom or characteristic (National Institute of Mental Health (NIMH), 2022) – therein implying a categorical perspective of mental health with one person having a mental illness, while the other three do not.

17. Second, the body of work by Georg Schomerus and others on the continuum model is entirely neglected; instead the hypotheses are presented as though no one had tested them before.

We have added references to the work of Georg Schomerus and colleagues within the introduction and used the latest work from this group to help put our study into context:

In a recent review, the opposing perspective of a non-categorical approach (also referred to as the continuum approach) was shown to generally be associated with a reduction in mental health stigma (Peter et al., 2021). But while this approach may be superior in reducing stigma, it may have some unintended negative consequences. There is some limited literature suggesting that over-normalising mental health problems (i.e. removing any notion of “otherness”) can adversely impact help-seeking (Biddle et al., 2007; Fernandez et al., 2022).

We also referred to this work in order to establish the rationale for our project:

The review by Peter et al. (2021) brought together the findings of 8 intervention studies, three of which were with members of the public. However, Peter et al. (2021) highlights these studies are limited in that they did not manipulate the intervention message and therefore cannot provide any causal evidence of their impact of non-categorical beliefs on stigma. To our knowledge, there is currently no experimental test of the impact of a categorical versus non-categorical anti-stigma campaign that assesses its impact on both stigma and help-seeking outcomes.

18. Methods: As this is an online RCT I recommend the use of the online RCT extension of the Consort reporting guidance, including a consort flow diagram.

Based on a search of the EQUATOR network guidelines, we have decided to use the CONSORT-SPI 2018 Extension guidelines as this is the best fit for experimental studies in the area of psychology. 

19. Method: The categorical poster description of diagnosis/problem appears to assume that these are synonymous in people’s minds. This is not the case- many people think of nondiagnostic issues as mental health problems. This is something that increased over Time to Change, the evaluation of which included assessment of whether people think of grief and stress as mental health problems. 

We have made some changes to the wording in how we describe the posters to make it clear that the categorical approach refers to there being a boundary delineating good and poor mental health, and that this may, but not necessarily, reflect the presence and absence of a psychiatric diagnosis:

The categorical poster aligns with the medical model of mental health whereby good and poor mental health can be clearly delineated with cut-offs that describe the person as either having or not having a diagnosable mental health problem (American Psychiatric Association, 2013; World Health Organisation (WHO), 1992)… The non-categorical poster moves away from cut-offs and/or diagnostic labels and instead emphasises that we all have mental health and that this is fluctuating and changeable.

20. Method: The AQ ask about people with mental health problems, implying a categorical attitude. Is this a problem for measurement of the impact of the noncategorical poster?

Thank you for raising this issue. We have acknowledged this within our limitations:

Further issues related to our method of measurement is that the wording of the AQ may have tainted our experimental manipulation – especially for the non-categorical poster arm. The language used in the AQ may have subtly communicated an alignment with the categorical approach i.e. referring to people with mental health problems, suggesting an “otherness”. Participants completing this questionnaire in the non-categorical arm will have viewed potentially contradictory ideas and the impact of this on their responses cannot be fully determined. Addressing this in future research studies will be a challenge for researchers who must consider ways of measuring endorsement of competing ideologies using neutral language.

21. Discussion: Given the evidence cited for the lack of impact of short-term campaigns on the outcomes of interest it is not clear what the rationale for the current study was.

We included additional text in the introduction that sets out the rationale for this project. 

The review by Peter et al. (2021) brought together the findings of eight intervention studies, three of which were with members of the public. However, Peter et al. (2021) highlights these studies are limited in that they did not manipulate the intervention message and therefore cannot provide any causal evidence of their impact of non-categorical beliefs on stigma. To our knowledge, there is currently no experimental test of the impact of a categorical versus non-categorical anti-stigma campaign that assesses its impact on both stigma and help-seeking outcomes.

We acknowledge that short term anti-stigma campaigns have limited effectiveness but as this type of study had not been done before (i.e. an experimental test of categorical versus non-categorical anti-stigma campaigns that assess the impact on both stigma and help-seeking outcomes) we felt that using brief campaigns was most appropriate for this initial test of our hypotheses. We acknowledge the issues with brief anti-stigma campaigns in our discussion. 

22. The implications section again ignores the existing body of work on the continuum model. How does the current study build on the work of Schomerus et al? Without such contextualisation and given the likely effect of a high prevalence message on reducing othering it is hard to see what this study adds.

We have amended the implications section of the discussion to make mention of the work of Schomerus and colleagues. 

Our null findings mean we cannot offer any suggestion as to which approach, categorical or non-categorical, is most effective at reducing mental health stigma and encouraging help-seeking when needed. Previous literature suggests a superiority of the non-categorical (continuum) approach, but these intervention studies are limited by their use of cross-sectional rather than experimental designs (Peter et al., 2021). We therefore do not yet know if there is an approach to anti-stigma campaigns that can surpass the gains achieved by the Time to Change approach.

We thank the reviewers for taking the time to review our paper and for offering constructive comments to strengthen it. Please do not hesitate to contact a member of our team if any of the points require further clarification.

Best wishes,

Dr Cassie M Hazell (on behalf of the wider research team).

---

## [Decision Letter · Decision Letter 1]

18 Jul 2022

PONE-D-22-07331R1Is it time to change the approach of mental health stigma campaigns? An experimental investigation of the effect of campaign wording on stigma and help-seeking intentions.PLOS ONE

Dear Dr. Hazell,

Thank you for submitting your manuscript to PLOS ONE. After careful consideration, we feel that it has merit but does not fully meet PLOS ONE’s publication criteria as it currently stands. Therefore, we invite you to submit a revised version of the manuscript that addresses the points raised during the review process.

We look forward to receiving your revised manuscript.

Kind regards,

Naoki Yoshinaga

Academic Editor

PLOS ONE

Journal Requirements:

Additional Editor Comments:

One of the reviewers suggested a minor revision. Please see the reviewer's comments.

Reviewers' comments:

Reviewer's Responses to Questions

**Comments to the Author**

1. If the authors have adequately addressed your comments raised in a previous round of review and you feel that this manuscript is now acceptable for publication, you may indicate that here to bypass the “Comments to the Author” section, enter your conflict of interest statement in the “Confidential to Editor” section, and submit your "Accept" recommendation.

Reviewer #1: All comments have been addressed

Reviewer #2: (No Response)

2. Is the manuscript technically sound, and do the data support the conclusions?

Reviewer #1: Yes

Reviewer #2: Partly

3. Has the statistical analysis been performed appropriately and rigorously? 

Reviewer #1: Yes

Reviewer #2: Yes

4. Have the authors made all data underlying the findings in their manuscript fully available?

Reviewer #1: Yes

Reviewer #2: Yes

5. Is the manuscript presented in an intelligible fashion and written in standard English?

Reviewer #1: Yes

Reviewer #2: Yes

6. Review Comments to the Author

Reviewer #1: I recommend it's publication. It is a well written article on an important topic. It was already of a high quality. The authors responded diligently to both my and the other reviewers comments. It was a pleasure to review.

Reviewer #2: Regarding the authors' response:

To our knowledge, there is currently no experimental test of the impact of a categorical versus non-categorical anti-stigma campaign that assesses its impact on both stigma and help-seeking outcomes.

The authors did not assess help seeking outcomes. The abstract begins accurately by using the term help seeking intentions, and then changes to help seeking outcomes. Please correct this. Once this is accurate, the rather minimal difference in outcomes used between this study and doi: 10.1016/j.eurpsy.2015.11.006 becomes apparent; the overemphasis of the novelty of the study then still needs to be addressed.

7. PLOS authors have the option to publish the peer review history of their article (what does this mean?). If published, this will include your full peer review and any attached files.

Reviewer #1: **Yes: **Daniel Walsh

Reviewer #2: **Yes: **Dr Claire Henderson

---

## [Author Response · Author response to Decision Letter 1]

3 Aug 2022

Manuscript Title: Is it time to change the approach of mental health stigma campaigns? An experimental investigation of the effect of campaign wording on stigma and help-seeking intentions.

We thank the reviewers for their thoughtful and important comments. We are pleased to have the opportunity to revise our paper in light of their feedback. We have made changes to our manuscript using tracked changes. Our response letter is formatted with the reviewers’ comments given in bold, our response in normal typeface, and where needed, extracts from the paper italicised.

REVIEWER 1 COMMENTS:

1. I recommend it's publication. It is a well written article on an important topic. It was already of a high quality. The authors responded diligently to both my and the other reviewer’s comments. It was a pleasure to review.

We would like to thank the reviewer for their kind comments about our paper. 

REVIEWER 2 COMMENTS:

2. Regarding the authors' response: “To our knowledge, there is currently no experimental test of the impact of a categorical versus non-categorical anti-stigma campaign that assesses its impact on both stigma and help-seeking outcomes.” The authors did not assess help seeking outcomes. The abstract begins accurately by using the term help seeking intentions, and then changes to help seeking outcomes. Please correct this. Once this is accurate, the rather minimal difference in outcomes used between this study and doi: 10.1016/j.eurpsy.2015.11.006 becomes apparent; the overemphasis of the novelty of the study then still needs to be addressed.

We respect the feedback from the reviewer and appreciate the opportunity to improve our paper. The use of ‘help-seeking outcomes’ to describe our dependent variable is inappropriate and we have revised all instance of this to ‘help-seeking intentions’. 

With regard to the novelty of our paper, we do agree that there are similarities with the paper by Schomerus et al. (2016). Both papers are comparing continuum versus dichotomous approaches to mental health on mental health on mental health stigma. We do however posit that our paper is distinct from the work by Schomerus et al. (2016) in that we also assessed help-seeking intentions. The paper by Schomerus et al. (2016) does not assess this construct (the measures included are the Social Distance Scale, continuum and difference beliefs, and stereotype endorsement). We would therefore like to request that the current rationale remains as presented in the revised version of the manuscript. 

We thank the reviewers for taking the time to review our paper and for offering constructive comments to strengthen it. Please do not hesitate to contact a member of our team if any of the points require further clarification.

Best wishes,

Dr Cassie M Hazell (on behalf of the wider research team).

---

## [Decision Letter · Decision Letter 2]

5 Aug 2022

Is it time to change the approach of mental health stigma campaigns? An experimental investigation of the effect of campaign wording on stigma and help-seeking intentions.

PONE-D-22-07331R2

Dear Dr. Hazell,

We’re pleased to inform you that your manuscript has been judged scientifically suitable for publication and will be formally accepted for publication once it meets all outstanding technical requirements.

Kind regards,

Naoki Yoshinaga

Academic Editor

PLOS ONE

Additional Editor Comments (optional):

Reviewers' comments:

Reviewer's Responses to Questions

**Comments to the Author**

1. If the authors have adequately addressed your comments raised in a previous round of review and you feel that this manuscript is now acceptable for publication, you may indicate that here to bypass the “Comments to the Author” section, enter your conflict of interest statement in the “Confidential to Editor” section, and submit your "Accept" recommendation.

Reviewer #2: All comments have been addressed

2. Is the manuscript technically sound, and do the data support the conclusions?

Reviewer #2: Yes

3. Has the statistical analysis been performed appropriately and rigorously? 

Reviewer #2: Yes

4. Have the authors made all data underlying the findings in their manuscript fully available?

Reviewer #2: Yes

5. Is the manuscript presented in an intelligible fashion and written in standard English?

Reviewer #2: Yes

6. Review Comments to the Author

Reviewer #2: no further comments.................................................................................

7. PLOS authors have the option to publish the peer review history of their article (what does this mean?). If published, this will include your full peer review and any attached files.

Reviewer #2: **Yes: **Claire Henderson

---

## [Editor Report · Acceptance letter]

9 Aug 2022

PONE-D-22-07331R2 

Is it time to change the approach of mental health stigma campaigns? An experimental investigation of the effect of campaign wording on stigma and help-seeking intentions. 

Dear Dr. Hazell:

I'm pleased to inform you that your manuscript has been deemed suitable for publication in PLOS ONE. Congratulations! Your manuscript is now with our production department. 

Kind regards, 

on behalf of

Prof. Naoki Yoshinaga 

Academic Editor

PLOS ONE